# Differential Cell Line Susceptibility to the SARS-CoV-2 Omicron BA.1.1 Variant of Concern

**DOI:** 10.3390/vaccines10111962

**Published:** 2022-11-18

**Authors:** Hitesh Dighe, Prasad Sarkale, Deepak Y. Patil, Sreelekshmy Mohandas, Anita M. Shete, Rima R. Sahay, Rajen Lakra, Savita Patil, Triparna Majumdar, Pranita Gawande, Jyoti Yemul, Pratiksha Vedpathak, Pragya D. Yadav

**Affiliations:** Indian Council of Medical Research-National Institute of Virology, Pune 411021, Maharashtra, India

**Keywords:** SARS-CoV-2, cell susceptibility, omicron, cytopathic effect, virus replication

## Abstract

The unique mutations of the SARS-CoV-2 Omicron variant are associated with increased transmissibility, immune escape, increased binding affinity to ACE-2, and increased viral load. Omicron exhibited a shift in tropism infecting the upper respiratory tract compared to other variants of concern which have tropism for the lower respiratory tract. The tropism of omicron variants in cell lines of different hosts and tissue origins still remains unclear. Considering this, we assessed the susceptibility of different cell lines to the SARS-CoV-2 omicron BA.1.1 variant and permissiveness among different cell lines for omicron replication. Susceptibility and permissiveness of a total of eleven cell lines, including six animal cell lines and five human cell lines for omicron BA.1.1 infection, were evaluated by infecting individual cell lines with omicron BA.1.1 isolate at a 0.1 multiplicity of infection. Virus replication was assessed by observation of cytopathic effects followed by viral load determination by real-time PCR assay and virus infectivity determination by TCID50 assay. The characteristic cytopathic effect, increased viral load, and productive omicron replication was detected in Vero CCL-81, Vero E6, Vero/hSLAM, MA-104, and Calu-3 cells. Although LLC MK-2 cells showed an increased TCID50 titer at the second infection, the viral load did not show much difference in both infections. Caco-2 cells did not show evident CPE, but they supported omicron replication at a low level. A549, RD, MRC-5, and BHK-21 cells supported omicron BA.1.1 replication without the CPE. This is the first study on the comparison of susceptibility of different cell lines to Omicron variant BA.1.1, which might be useful for future studies on emerging SARS-CoV-2 variants.

## 1. Introduction

The novel beta coronavirus called Severe acute respiratory syndrome coronavirus-2 was identified for the first time in Wuhan, China, in late 2019 [1,2]. Within months of its first emergence, it has spread to various countries of the globe, with the World Health Organization (WHO) declaring it as a pandemic in March 2020. As of 16 September 2022, more than 603 million people have been infected with this rapidly mutating virus, with more than 6 million deaths [3]. The SARS-CoV-2 genome rapidly acquired newer mutations, and new variants are being identified with increased fitness, transmissibility, and pathogenic properties from late 2020 [4]. Various Variants of Concern (VOCs) and/or Variants under Investigation (VUIs) arise with newer characteristics. The first VOC designated Alpha (B.1.1.7) detected in the United Kingdom showed higher transmissibility than the Wuhan strain. However, it is susceptible to neutralization by the immune system [5]. Shortly after the Alpha variant, new VOCs, namely Beta (B.1.351) and Gamma (P.1), emerged and showed antibody resistance by escaping from neutralizing antibodies [6,7]. Shortly after these, a highly pathogenic variant designated Delta (B.1.617.2), emerged in India with high pathogenicity and immune escape and was associated with a deadly second COVID-19 wave in India [8,9]. In November 2021, a new variant designated, Omicron (B.1.1.529) emerged in Africa, which contains at least 30 amino acid substitutions in the spike protein alone, including 15 mutations in the receptor-binding domain (RBD) of spike protein [10,11]. It has evolved enormously into BA.1, BA.2, BA.3, BA.4, BA.5 lineage, and more than 300 sub-lineages. All lineages are currently circulating worldwide with significant disease burdens despite successful vaccination [12]. In India, an intense third COVID-19 wave occurred due to the omicron variant despite a massive vaccination drive, suggesting the variant’s vaccine escape potential and rapid transmissibility [13].

In the face of rapid and massive vaccination campaigns worldwide, the rapidly evolving SARS-CoV-2 is infecting unvaccinated, partially vaccinated as well as fully vaccinated individuals by evading host immune responses with less severity [7,8,10]. Considering this, it becomes essential to isolate these newly emerged variants in the cell culture system for studying virus characteristics (replication, tropism, pathogenicity, etc.), designing potential drugs, testing antiviral compounds, vaccine development, and studying vaccine effectiveness [14,15].

Cell line susceptibility of SARS-CoV-2 depends on cell tropism, receptor expression levels, virus replication kinetics, and clinical and epidemiological characteristics of SARS-CoV-2 [16]. SARS-CoV-2 uses angiotensin-converting enzyme 2 (ACE2) as a cellular entry receptor [17]. Apart from ACE2, SARS-CoV-2 uses Transmembrane protease serine 2 (TMPRSS2) molecules, a protease required for efficient entry into human cells [18]. In cells lacking TMPRSS2, cellular entry of SARS-CoV-2 is mediated by cathepsin [16,19]. The virus uses the Receptor-binding domain (RBD) domain of its spike protein to attach to host cellular ACE2 receptors [17]. The African Green Monkey-derived Vero E6 kidney cell line is a widely used cell line for SARS-CoV-2 virus stock propagation and antiviral assays [20,21]. SARS-CoV-2 entry in Vero E6 cells is cathepsin-mediated due to the lack of cell surface TMPRSS molecules [15,20].Cell lines expressing both ACE2 and TMPRSS2 are highly permissive for SARS-CoV-2 infection and have been employed in virus characterization [15,17,21]. SARS-CoV-2 replicates with high permissivity in human colon epithelial carcinoma cell line Caco-2 and human airway epithelial carcinoma cell line Calu-3 [22,23]. Neutralizing antibodies against the RBD of spike protein have proved to be effective against SARS-CoV-2 [5]. The continuous emergence of SARS-CoV-2 variants with potential mutations, particularly in the RBD of spike protein, is the cause of vaccine escape [6,7,10]. Moreover, it has been found that the recently emerged Omicron variant replicates less efficiently in TMPRSS2-expressing cell lines when compared to the Delta variant and omicron uses an endocytosis pathway for entry in these cell lines [20]. Hence, the emergence of highly mutated SARS-CoV-2 variants demands identifying the susceptibility of the virus to ACE2 and TMPRSS2 receptor expression levels in various cell lines, as the current vaccines are not able to give complete protection against emerging new variants and the scope for antiviral treatment in people affected by COVID-19 is still unclear. Thereby, continuous studies of the susceptibility of various cell lines are important to re-evaluate the utility and responses of the various cell lines in vitro and evaluate candidate therapeutics against SARS-CoV-2.

Foreseeing this, we characterized the susceptibility of various cell lines, including five human cell lines derived from the lung, muscle, and colon carcinoma; four African green monkey kidney-derived cell lines; a cell line derived from the kidney of *Macaca mulatta* and a cell line derived from hamster kidney cells to SARS CoV-2 Omicron B.1.1 variant. Viral load in culture supernatants, virus growth properties, cell morphology changes, and infectious progeny virion yield were assessed following omicron infection.

## 2. Materials and Methods

### 2.1. Virus Propagation and Titration

SARS-CoV-2 Omicron (B.1.1) isolate (MCL-21-H-11828, GSAID accession no. EPI_ISL_8542931) used in this study was originally isolated from the human nasopharyngeal/oropharyngeal swab in Vero CCL-81 cell culture [24]. Briefly, Vero CCL-81 cells grown on T-175cm^2^ TC treated flasks (Eppendorf, Hamberg, Germany) were infected with the isolate at a multiplicity of infection (MOI) of 0.1. The cells were observed daily for cytopathic effect. Three days post-infection, Vero CCL-81 cells showed evident CPE in the form of cell rounding and detachment. Flasks were freezed on three days post-infection when cells were showing 4+ CPE at −86 °C until further use. The tissue culture fluid (TCF) obtained was centrifuged at 3000 rpm (revolutions per minute) for 5 min at 4 °C, the supernatant was harvested, and the virus titer was determined by end-point dilution method. The 50% tissue culture infective doses (TCID 50) were calculated on the 5th day by the Reed and Muench method [25]. The titered virus was used for further susceptibility studies. All the experiment involving live SARS-CoV-2 was performed in the Biosafety level 4 laboratory at ICMR-National Institute of Virology, Pune, India.

### 2.2. Cell Culture Systems

A total of eleven cell lines, i.e., African green monkey kidney clone E6 (Vero E6) cells, Vero CCL-81 cells, African green monkey embryo with clone MA-104 (MA-104) cells, human rhabdomyosarcoma (RD) cells, Homo sapiens-lung (MRC-5) cells, Human colorectal adenocarcinoma (Caco-2) cells, Human lung adenocarcinoma (Calu-3) cells, Baby hamster kidney (BHK-21), Vero/hSLAM, A-549, and LLC-MK-2 were selected to test the susceptibility of omicron variant. All the cell lines were obtained from ATCC and maintained in the cell culture laboratory. The cell lines were also tested for mycoplasma using the EZdetect^TM^PCR kit for mycoplasma detection (HiMedia, Mumbai) and were found to be negative for mycoplasma.

All the cell lines were individually grown in their respective growth medium and culture conditions (Table 1).Vero E6 cells, Vero CCL-81 cells, MA-104 cells, RD cells, MRC-5 cells, Caco-2 cells, Calu-3 cells, and BHK-21 cells were grown in Eagle’s minimum essential medium (MEM) (HiMedia, Mumbai) supplemented with 10% FBS (HiMedia, Mumbai), penicillin (100 U/mL) and streptomycin (100 mg/mL). Vero/hSLAM cells were grown in DMEM (HiMedia, Mumbai) supplemented with 10% FBS, penicillin (100 U/mL), and streptomycin (100 mg/mL). A-549 cells were grown in Nutrient mixture F-12 K HAM (HiMedia, Mumbai) supplemented with 10% FBS penicillin (100 U/mL) and streptomycin (100 mg/mL),and LLC-MK-2 cells were grown in Medium 199 (HiMedia, Mumbai) with 1 % Horse serum (HiMedia, Mumbai), penicillin (100 U/mL) and streptomycin (100 mg/mL). All cell lines were grown at 37 °C with 5% CO_2_ and 75–80% humidity in a humidified CO_2_ incubator.

### 2.3. Virus Infectivity Assay

Individual cells were seeded in 24-well plates and incubated till they reached 60–80% confluency at the start of the infection assay. The assay was performed in duplicate for all the cell lines under study. Briefly, for the first round of infection (P-1), the growth medium was discarded, and the cells were infected with 100 µL of 0.1 MOI of SARS-CoV-2 Omicron (BA.1.1) isolate. Cells were incubated at 37 °C with 5% CO_2_ for a viral adsorption period of 1 h with intermittent shaking of plates at regular intervals. After adsorption time, the inoculum was discarded; cells were washed with 1X PBS (HiMedia, Mumbai) and supplemented with appropriate fresh medium containing 2% serum (Details mentioned in Table 1). The cells were incubated at 37 °C in a 5% CO_2_ atmosphere for a period of 5 days and were observed daily for any cellular changes using an inverted microscope (Nikon, Eclipse Ti, Japan). On 5 days post-infection, the plates were frozen at −86 °C until further use. For the second round of infection (P-2), P-1 plates were quickly thawed; the TCF was centrifuged at 3000 rpm for 5min, and the supernatant was harvested. 100 µL of P-1 supernatant was used as inoculum to infect fresh cells for the second round of infection (P-2). The rest of the procedure was similar to the first round of infection. The supernatant of all cell lines from both rounds of infection was subjected to SARS-CoV-2-specific real-time RT-PCR. Median tissue culture infectious dose 50 (TCID50/mL) was obtained for all the culture supernatants harvested at the 5th PID by Reed and Muench method [25].

### 2.4. RNA Extraction and Real-Time Reverse-Transcription PCR

Total RNA was extracted from the culture supernatants of all 11 cell lines studied at each round of infection with an automated RNA extraction system using Magmax RNA Extraction Kit (Applied Biosystems, Waltham, MA, USA) according to the manufacturer’s instructions. Briefly, 200 µL of supernatant was used to extract total nucleic acid (TNA) eluted in 60 µL elution buffer. The RNA extracted was used for the real-time quantitative reverse-transcription polymerase chain reaction (rRT-PCR) for the quantitative detection of the E gene, the RDRP2 gene, and E and N sub-genomic RNA of SARS-CoV-2. A series of 10-fold dilutions equivalent to 1 × 10^2^–1 × 10^6^ copies/reaction mixture were prepared to generate a standard curve and run in parallel with the test samples to calculate the viral copy number. The viral RNA copies of the inoculum inoculated onto cells for the first round of infection were defined as a baseline viral load. In order to evaluate the efficiency (permissiveness) of virus proliferation in each cell line, viral loads in P-1 and P-2 culture supernatants from infected cells were normalized to the baseline as fold changes and values correlated.

### 2.5. Statistical Analysis

The TCID50 and viral load values of P1 and P2 infection for the different cell lines were analyzed using paired *t* test with GraphPad Prism software version 8.4.3 (GraphPad, San Diego, CA, USA). The *p*-values less than 0.05 were considered significant.

## 3. Results

We studied a total of eleven cell lines (Table 1) to check for their susceptibility to the SARS-CoV-2 Omicron (BA.1.1) variant. The Omicron (BA.1.1) variant tropism for the studied cell lines was determined by observation of CPE, viral load, and virus titer determination.

### 3.1. Virus Propagation and Titration

The Omicron (BA.1.1) isolate used in the study showed a characteristic rounding and detachment of cells from the flask surface starting from day 2 post-infection. On day 3 post-infection, 4+ CPE (was observed, with the detachment of cell monolayer from the surface of tissue culture flasks and the refractive appearance of infected cells. The 50% tissue culture infective dose (TCID 50) was found to be 10^5.3^/mL.

### 3.2. Determination of Virus Infectivity Based on Cytopathic Effect

Of the eleven cell lines studied, CPE was observed for Vero CCL-81, Vero E6, Vero/hSLAM, MA-104, LLC MK-2, and Calu-3 cells in both P-1 and P-2. No visible CPE or any other morphological changes were observed for A549, RD, MRC-5, Caco-2, and BHK-21 cells in both P-1 and P-2.The morphological changes in Vero CCL-81 and Vero E6 cells started to appear on the 2nd PID. Vero CCL-81 cells showed distinct morphological changes with the rounding of cells, the refractive appearance of cells, and the sloughing of cells throughout the cell culture flask. A characteristic CPE with cell rounding, and clumping of detached cells was observed in Vero E6 cells. The Vero/hSLAM cells showed morphological changes on the 3rd PID with bulging and clumping of detached cells. Slight morphological changes without evident CPE were observed for MA-104 cells starting from the 4th PID in P-1 and from the 3rd PID in P-2. The signs of slight morphological changes, including cell detachment, and cell clumping without any obvious CPE, were observed in LLC MK-2 cells at P-1 and P-2.

There were no morphological changes observed for Caco-2 cells, and cells remained healthy with no apparent cell damage and kept on dividing. Calu-3 cells showed morphological changes with cells bulging and rounding starting from the 4th PID in the first round of infection and from the 3rd PID in the second round of infection. No cytopathic changes were observed for A549, RD, MRC-5, and BHK-21 cells in both the first and second rounds of infection (Figure 1 & Table 2).

### 3.3. Determination of Virus Infectivity Based on Viral Titer

Growth properties of Omicron (BA.1.1) were subsequently characterized by determining infectious progeny virus yields as TCID50/mL in debris-clarified supernatants collected at 5th PID at both rounds of infection, and virus titer values were correlated with viral load. Of all the cell lines studied, Vero CCL-81 cells generated higher viral titers of 10^5.33^ TCID50/mL, compared with a low titer of 10^2.67^ TCID50/mL for MA-104 and LLC MK-2 cells and the lowest viral titer of 10^2.5^ TCID50/mL for Vero/hSLAM cells in P-1 supernatants (Table 3). Vero E6 and Calu-3 cells showed viral titers of 10^4.67^ TCID50/mL and 10^3^ TCID50/mL, respectively, in P-1 supernatants, suggesting their considerable permissiveness for the Omicron (BA.1.1) variant. There was a slight increase in viral titers for Vero CCL-81, MA-104, LLC MK-2, and Calu-3 cells in P-2 supernatants when compared with their respective P-1 viral titers (Table 3). Vero E6, Vero/hSLAM, and Caco-2 cells showed similar TCID50/mL virus titers of 10^4.67^, 10^2.5^, and 10^3^, respectively, in both P-1 and P-2. The study shows that Vero CCL-81 cells have the highest viral titer of 10^5.67^ TCID50/mL of all the cell lines studied, suggesting its high permissiveness for the Omicron (BA.1.1) variant. Despite the highest viral loads of 1.0 × 10^11^ copies/mL and 1.4 × 10^11^ copies/mL in P-1 and P-2, respectively, in MA-104 cells, MA-104 cells showed low virus titer (10^2.67^ TCID50/mL and 10^3.67^ TCID50/mL in P-1 and P-2 respectively).

The absence of cytopathic changes in A549, MRC-5, and BHK-21 cells suggests that these cell lines could not be used for a variety of research purposes as there could not be a known viral titer. Despite a slight increase in viral titer in P-2 for RD cells, no TCID50 titer could be determined for these cells, indicating their non-permissiveness. Caco-2 cells also didn’t show any cellular changes but showed productive replication in P-1 as well as virus titer of 10^3^ TCID50/mL in P-1 and P-2, suggesting their permissivity for the virus, albeit at a low level.

### 3.4. Determination of Virus Infectivity Based on the Viral Load

Viral loads varying from 6.1× 10^8^ to 1.0 × 10^11^ copies/mL were detected in culture supernatants of P-1 of all the cell lines. Whereas viral loads for P-2 culture supernatants for all the cell lines varied from 1.8 × 10^7^ to 1.4 × 10^11^ copies/mL (Table 3). To compare the efficiency of virus replication with the passage as well as between different cell lines, we defined initial viral RNA copies used to infect P-1 cells as a viral load baseline. Viral load values of each cell line in both P-1 and P-2 were normalized against a baseline viral load of 2.9 × 10^10^ copies/mL.

The P-1 viral load levels for Vero CCL-81, Vero E6, Vero/hSLAM, MA-104, LLC MK-2, Calu-2, and Calu-3 cells were 0.17–3.48 times more than the baseline viral load, suggesting virus replication in these cells. P-2 was more productive when compared to P-1 for Vero CCL-81, Vero E6, Vero/hSLAM, MA-104, and Calu-3 cells, as evidenced by an increase in viral load with 0.97–4.96 fold change of viral load suggesting sustainable viral production in these cells.

The viral loads in MA-104 cells were significantly higher compared to other cell lines in both P-1 and P-2. The viral load in the P-2 supernatant (8.9 × 10^10^ copies/mL) of Calu-3 cells increased significantly when compared with the P-1 viral load of 2.4 × 10^10^ copies/mL, suggesting increasing permissiveness of Calu-3 cells with the passage. LLC MK-2 cells had a varied viral load in P-2 (4.4 × 10^9^ copies/mL) compared with P-1 viral load (1.6 × 10^10^ copies/mL) with normalized viral load decreasing from 0.55 in P-1 to 0.15 in P-2, suggesting their low permissiveness. The RD cells showed increased viral load in P-2 (5.76×10^9^ copies/mL) compared to the P-1 viral load of 2.55 × 10^9^ copies/mL. The viral load in A549, MRC-5, and BHK-21 cells decreased in P-2 when compared to the P-1 viral load by 0.02, 0.05, and 0.02 times respectively.

To verify productive virus proliferation with the passage as well as to confirm cell susceptibility of cell lines not showing evident CPE, supernatants from both rounds of infection from all cell lines were subjected to TCID50 assay performed on Vero CCL-81 cells.

Based on the observation of CPE, viral load in culture supernatants, and infectious progeny virion titer determined at each passage in Vero CCL-81 cells, we defined SARS-CoV-2 Omicron BA.1.1-permissive cell lines as having viral loads of fold changes of >0.15 in addition to TCID50 titer in Vero CCL-81 cells.

### 3.5. Statistical Analysis

The comparison of TCID 50 values in P1 and P2 infection was found to be insignificant for monkey cell lines (*p* value = 0.18) and human cell lines (*p* value = 0.37). The viral load values of monkey cell lines were significant in P1 and P2 infection (*p* value = 0.02), while it was insignificant for human cell lines (*p* value = 0.39).

## 4. Discussion

Since early 2020, SARS-CoV-2 has been causing a devastating pandemic with more than 6 million deaths worldwide. In spite of rapid and massive vaccination worldwide, breakthrough infections are common due to rapidly mutating SARS-CoV-2 with improved virus fitness. Knowledge about the change in host/cell tropism, transmissibility rate, pathogenicity, and immune escape potential of newly emerged variants is very limited. Vaccine escape of SARS-CoV-2 variants is mainly attributed to mutations in the Receptor-binding domain (RBD) of the spike protein. Mutations in the RBD region potentially impart changes in cell tropism, host tropism, or efficient entry into host cells, as well as being responsible for vaccine escape.

SARS-CoV-2 entry into host cells is by ACE-2-mediated endocytosis as well as by cathepsin-mediated endocytosis [16,17].TMPRSS2 proteases often aid ACE2 molecules for more efficient entry [18]. Cellular entry of strains of SARS-CoV-2 circulating during 2020 till mid-2021, including Alpha, Beta, Gamma, and Delta variants, was mostly enhanced by TMPRSS2 [16,17,18,23]. Studies have found that the Omicron variant replicates less efficiently in TMPRSS2-expressing cells, and its entry is primarily by cathepsin-mediated endocytosis [23]. This change in the molecular tropism of omicron variants and their further derivatives highlight the need to identify the susceptibility of emerging variants to ACE2 and TMPRSS2 molecules as well as in various cell lines. It becomes crucial to isolate the variants as early as possible, studying their susceptibility in various cell lines; evaluate variant characteristics to reevaluate the utility of SARS-CoV-2 study platforms.

Since the start of the SARS-CoV-2 pandemic, the interferon-deficient Vero cells of the African green monkey kidney cell origin are the cells of choice for rapid isolation of the virus, screening antiviral compounds, and neutralization assays. Several studies have demonstrated that cell lines of monkey kidney origin, such as Vero CCL-81, Vero E6, Vero/hSLAM, MA-104, and LLC MK-2, are susceptible to SARS-CoV-2 infection [25,27]. Our study reaffirms these findings, particularly focusing on the recently emerged omicron variant, as well as, evaluating, and comparing the permissiveness of the Omicron variant in each cell line. Our study complements recent findings by Pawar et al. of isolation of SARS-CoV-2 in Vero/hSLAM and MA-104 while attempting to isolate Measles, Rubella, and Rotavirus [28]. All the cell lines used in this study i.e., Vero CCL-81, Vero E6, Vero/hSLAM, MA-104, LLC MK-2, A549, RD, MRC-5, Caco-2, Calu-3 have been used in SARS-CoV-2 research except RD cell line [14,15,18,26].

In order to obtain a clear picture of SARS-CoV-2 infectivity and pathogenesis; it is essential to search for cell lines of human origin susceptible to SARS-CoV-2. Human cell line platforms are more reliable for screening antivirals, developing of new therapeutics, and production of viruses in bulk for vaccine research. In this study, we characterized CPE, viral load, infectious progeny virion yield, and variant analysis in 11 cell lines with Omicron (BA.1.1) variant.

Our study evaluates the susceptibility of the omicron variant in diverse tissues of human origin, including lung epithelia, lung fibroblast, lung epithelial carcinoma, muscle carcinoma, and colon epithelial carcinoma. Our study shows that three cell lines of human origin, A549, RD and MRC-5 were susceptible to BA.1.1 infection without the evident CPE. Our study has found that SARS-CoV-2 efficiently replicates in human airway epithelial carcinoma, Calu-3 cells, whereas it replicates less efficiently in colon epithelial carcinoma Caco-2 cells, complementing the previous research [27]. Unlike the evident and characteristics of CPE observed in Caco-2 cells upon infection with SARS-CoV-1, Caco-2 cells do not show any evident CPE upon infection with SARS-CoV-2, but they are susceptible and permissive for SARS-CoV-2 at a low level [26]. Our recent findings suggest the permissivity of baby hamster kidney cells (BHK-21) for omicron infection without evident CPE.

This study compared the susceptibility of the SARS-CoV-2 Omicron (BA.1.1) variant in human and animal cell lines derived from different tissues. Our findings provide an important reference for the use of cell lines for SARS-CoV-2 studies. In conclusion, it is necessary to continuously evaluate cell susceptibility with newly emerging SARS-CoV-2 variants for future translational research, i.e., the development of vaccines and therapeutics.

The only limitation of this study is that we carried out a cell susceptibility study only with the omicron (BA.1.1). As the omicron has evolved into five lineages, i.e., BA.1, BA.2, BA.3, BA.4, BA. 5 and more than 300 sub-lineages, it is difficult to carry out the such time-consuming studies even with specific lineages of Omicron.

## Figures and Tables

**Figure 1 vaccines-10-01962-f001:**
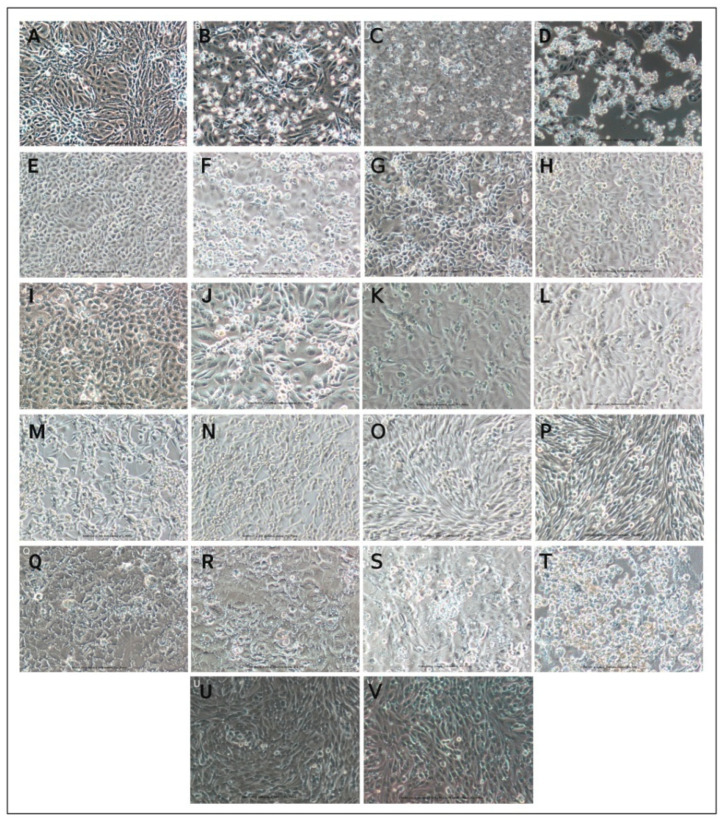
Cell lines were inoculated with the SARS-CoV-2 Omicron (BA.1.1) variant, and uninfected cells were used as a cell control for each cell line. Microscopy images (original magnification ×10) captured on 5 days post-infection. The cytopathic effect of omicron BA.1.1 variant in different cell lines. (**A**) Uninfected Vero CCL-81 cells; (**B**) Infected Vero CCL-81 cells; (**C**) Uninfected Vero E6 cells; (**D**) Infected Vero E6 cells; (**E**) Uninfected Vero hSLAM cells; (**F**) Infected Vero/hSLAM cells; (**G**) Uninfected MA-104 cells; (**H**) Infected MA-104 cells; (**I**) Uninfected LLC MK-2 cells PID5 P-2 (**J**) Infected LLC MK-2 cells; (**K**) Uninfected A549 cells; (**L**) Infected A549 cells; (**M**) Uninfected RD cells; (**N**) Infected RD cells; (**O**) Uninfected MRC-5 cells; (**P**) Infected MRC-5 cells; (**Q**) Uninfected Caco-2 cells; (**R**) Infected Caco-2 cells; (**S**) Uninfected Calu-3 cells; (**T**) Infected Calu-3 cells; (**U**) Uninfected BHK-21 cells; (**V**) Infected BHK-21 cells.

**Table 1 vaccines-10-01962-t001:** List of cell lines tested for susceptibility studies with SARS-CoV-2 Omicron BA.1.1 variant.

Sr. No.	Cell Lines	Source	Cell Type	Tissue	Culture Medium	Culture Conditions	References
**Monkey**							
1	Vero CCL-81	Vero (ATCC CCL-81)	Epithelial	Kidney	MEM + 10% FBS	37 °C, 5% CO_2_	[15,26]
2	Vero E6	ATCC no. CRL-1586	Epithelial	Kidney	MEM + 10% FBS	37 °C, 5% CO_2_	[14,15,18,26]
3	Vero/hSLAM	04091501-1VL	Epithelial	Kidney	DMEM + 10% FBS	37 °C, 5% CO_2_	[26]
4	MA-104	ATCC no. CRL-2378	Epithelial	Kidney	MEM + 10% FBS	37 °C, 5% CO_2_	[26]
5	LLC MK-2	ATCC no. CCL-7	Epithelial	Kidney	Medium 199 + 1% Horse serum	37 °C, 5% CO_2_	[15,26]
**Human**							
6	A549	ATCC no. CCL-185	Epithelial-like	Lung adenocarcinoma	Nutrient mixture F-12 k HAM	37 °C, 5% CO_2_	[14,15,18]
7	RD	ATCC no. CCL-136	Rhabdomyosarcoma	Muscle	MEM + 10% FBS	37 °C, 5% CO_2_	[Not available]
8	MRC-5	ATCC no. CCL-171	Fibroblast	Lung	MEM + 10% FBS	37 °C, 5% CO_2_	[26]
9	Caco-2	ATCC no. HTB-37	Epithelial	Colon cancer	MEM + 10% FBS	37 °C, 5% CO_2_	[15,26]
10	Calu-3	ATCC no. HTB-55	Epithelial	Lung adenocarcinoma	MEM + 10% FBS	37 °C, 5% CO_2_	[14,15]
**Hamster**							
11	BHK-21	ATCC no. CCL-10	Fibroblast	Kidney	MEM + 10% FBS	37 °C, 5% CO_2_	[15,26]

**Table 2 vaccines-10-01962-t002:** Differential cell line susceptibility to Omicron BA.1.1 variant as defined by cytopathic effect (CPE) on days 1 to 5 after infection with SARS-CoV-2 Omicron BA.1.1.

Sr. No.	Cell Line	First Round of Infection (P-1)					Second Round of Infection (P-2)				
		Day 1	Day 2	Day 3	Day 4	Day 5	Day 1	Day 2	Day 3	Day 4	Day 5
1	Vero CCL-81	N	1+	2+	4+	4+	N	1+	2+	4+	4+
2	Vero E6	N	1+	2+	4+	4+	N	1+	2+	4+	4+
3	Vero/hSLAM	N	N	1+	2+	4+	N	N	1+	2+	4+
4	MA-104	N	N	N	2+	4+	N	N	1+	3+	4+
5	LLC MK-2	N	N	N	1+	3+	N	N	1+	3+	4+
6	A549	N	N	N	N	N	N	N	N	N	N
7	RD	N	N	N	N	N	N	N	N	N	N
8	MRC-5	N	N	N	N	N	N	N	N	N	N
9	Caco-2	N	N	N	N	N	N	N	N	N	N
10	Calu-3	N	N	N	2+	4+	N	N	1+	2+	4+
11	BHK-21	N	N	N	N	N	N	N	N	N	N

N: No CPE, 1+: CPE (25% cells showing CPE), 2+: CPE (>25–50% cells showing CPE); 3+: CPE (>50–75% cells showing CPE), 4+: CPE (>75–100% cells showing CPE).

**Table 3 vaccines-10-01962-t003:** Cell lines tested for SARS-CoV-2 Omicron BA.1.1 variant infection susceptibility.

Cell Line	First Round of Infection (P-1)					Second Round of Infection (P-2)					Permissiveness (Yes/No)
	CPE Observation	Day of Start of CPE (PID)	TCID50/mL	Viral Loads (Copies/mL)	Times of Viral Loads Normalized to Baseline	CPE Observation	Day of Start of CPE (PID)	TCID50/mL	Viral Loads (Copies/mL)	Times of Viral Loads Normalized to Baseline	
**Monkey**											
Vero CCL-81	CPE Observed	2	10^5.33^	1.96 × 10^10^	0.675	CPE Observed	2	10^5.66^	2.83 × 10^10^	0.972	Highly permissive
Vero E6	CPE Observed	2	10^4.67^	1.53 × 10^10^	0.527	CPE Observed	2	10^4.67^	3.72 × 10^10^	1.281	Highly permissive
Vero/hSLAM	CPE Observed	3	10^2.5^	2.50 × 10^10^	0.859	CPE Observed	3	10^2.5^	3.18 × 10^10^	1.094	Permissive
MA-104	CPE Observed	4	10^2.67^	1.01 × 10^11^	3.482	CPE Observed	3	10^3.67^	1.36 × 10^11^	4.691	Permissive
LLC MK-2	CPE Observed	4	10^2.67^	1.59 × 10^10^	0.548	CPE Observed	3	10^3.23^	4.35 × 10^9^	0.15	Permissive
**Human**											
A549	No CPE	NA	0	1.65 × 10^9^	0.057	No CPE	NA	0	3.64 × 10^7^	0.001	Permissive
RD	No CPE	NA	0	2.55 × 10^9^	0.088	No CPE	NA	0	5.76 × 10^9^	0.198	Permissive
MRC-5	No CPE	NA	0	6.12 × 10^8^	0.021	No CPE	NA	0	2.70 × 10^7^	0.001	Permissive
Caco-2	No CPE	NA	10^3^	5.04 × 10^9^	0.173	No CPE	NA	10^3^	2.06 × 10^9^	0.071	Permissive
Calu-3	CPE Observed	4	10^3.67^	2.40 × 10^10^	0.826	CPE Observed	3	10^4.44^	8.90 × 10^10^	3.06	Highly permissive
**Hamster**											
BHK-21	No CPE	NA	0	1.26 × 10^9^	0.043	No CPE	NA	0	1.77 × 10^7^	0.001	Permissive

CPE: Cytopathic effect, PID: Post infection day; NA: Not applicable.

## Data Availability

All the data is included in the manuscript, and the GISAID identifier has been provided for the SARS-CoV-2 isolate.

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
