# Peer review of "Differential Cell Line Susceptibility to the SARS-CoV-2 Omicron BA.1.1 Variant of Concern"

_vaccines, 2022, doi:10.3390/vaccines10111962_

Round 1
Reviewer 1 Report
The goals of the authors are clear and the manuscript is well written, but the novelty of the work is little. Several publications have already compared the sensitivity of cell lines to SARS-Cov-2. doi: 10.1007/s10096-020-04106-0, 10.3201/eid2705.210023, 10.3390/v13122434. Regarding the most important cell lines examined by the authors, the results have already been published for the Omicron 2 BA.1.1 variant. 10.1186/s12985-022-01802-5, 10.3390/v14071369, 10.1080/22221751.2021.2023329.
The description of the experiments is incomplete, for example, the number of parallel experiments was not indicated. This makes it very difficult to evaluate the results in the current form of the manuscript.
Author Response
Reply to Reviewers Comments
The authors would like to express sincere thanks to the reviewers of Vaccines journal for vital comments on this manuscript for further improvement. As per the reviewer’s comments, the manuscript has been modified and the changes are presented in track-change mode. All the co-authors have participated in the revision of the manuscript.
Reviewer: 1
Comments to the Author (blind)
- The goals of the authors are clear and the manuscript is well written, but the novelty of the work is little. Several publications have already compared the sensitivity of cell lines to SARS-Cov-2. doi: 10.1007/s10096-020-04106-0, 10.3201/eid2705.210023, 10.3390/v13122434. Regarding the most important cell lines examined by the authors, the results have already been published for the Omicron 2 BA.1.1 variant. 10.1186/s12985-022-01802-5, 10.3390/v14071369, 10.1080/22221751.2021.2023329.
Reply:
- This study was done immediately after omicron variant detected in India (December-January 2022). As described in our research paper, it was difficult to isolate omicron variant directly from human clinical samples, so to isolate omicron variant, a combination of both in vivo and in vitro approaches were used. The difficulties in isolation of omicron variant in Vero cells necessitated to search for cell lines from diverse tissues and animal origins then available in our laboratory for rapid and easier isolation of omicron variant and this in turn helped in evaluate/ re-evaluate the utility of different cell lines for standardization of assays such as neutralization assay, antiviral assays, etc in the context of emerging omicron variants.
- Yadav PD, Gupta N, Potdar V, Mohandas S, Sahay RR, Sarkale P, Shete AM, Razdan A, Patil DY, Nyayanit DA, Joshi Y. Isolation and genomic characterization of SARS-CoV-2 omicron variant obtained from human clinical specimens. Viruses. 2022;14(3):461.
- The main objective of the study was to identify highly permissive cell lines for omicron replication so as to produce virus in bulk for future omicron tissue culture based booster vaccines.
- Recently, there was an increase in the COVID-19 cases with Omicron variants which has greater transmissibility and vaccine efficacy remarkably dropped. Although several publications has compared the sensitivity of cell lines to SARS-CoV-2; there is need to study the susceptibility of these cell lines with Omicron variants.
- The description of the experiments is incomplete, for example, the number of parallel experiments was not indicated. This makes it very difficult to evaluate the results in the current form of the manuscript.
Reply: The experiment was done in duplicate for each cell line and at each round of infection. The same has been added in the revised manuscript.
Reviewer 2 Report
Minor comments:
1. Editorial help with typos and formal English is warranted. Few examples: line 107 should read “on day 5”, line 109 “Biosafety level 4” no dashes, line 113: African green monkey embryo with clone MA-104; line 131: 5% CO2 and 75-80% humidity in a humidified CO2 incubator; line 154-155, word “using’ is used 3 times in one sentence.
2. Line 175: not sure what this statement means: “4+ CPE (75-100% cells showing CPE) was observed with enhanced CPE”
3. What was the exact difference in CPE between Vero CCL-81, Vero E6 and Vero/ hSLAM and other cells? “Vero CCL-81, Vero E6 and Vero/ hSLAM cells showed distinct morphological changes with rounding and sloughing of cells throughout the cell culture flask” but the following text is confusing as it uses the same term to describe CPE for other cells.
4. Figure 2 is redundant to the Table 3. Can be omitted.
Major points:
1. The premise of the testing these many different cell lines is not clearly articulated. Is it needed for neutralization assays? Drug testing? What was done before? How is this study compared to the previously done in the field and cell lines that are commonly used in the field?
2. If authors did not test Wuhan strain on these cell lines, it will be beneficial to add appropriate citations to the Table 1 and create an additional column so readers can find relevant information in the literature. If authors tested Wuhan strain on some or all of these cell lines, it will very useful to include that information also. Otherwise, there is an impression that some fo the cell lines that authors tested in the study very reported susceptible to Wuhan strain.
3. Approach to define permissiveness is questionable. First of all, it is defined much later in the text then it is actually used. Then, change by 0.187 units defines permissive vs highly permissive but 4.6-0.15 is still permissive although it is logical to assign 0.15 as lowly permissive.
4. Defining A549, RD and MRC5 as non-permissive is highly questionable. Obviously, there was no CPE on these cell lines, but virus was detected and it was infectious virus es evident by the productive P2 round. Moreover, differences in Baseline vs P1 and P1 vs P2 suggest that P2 was much more productive than P1 as there was not much less virus harvested after P2 compared to what was inoculated while during P1 the drop was significant. Therefore, qPCR indicates that these cell lines permitted virus infection without CPE. Because TCID50 measurement is based on CPE it yields obviously 0. To verify productive infection, it can be suggested to perform immunostaining of infected cells with antibody or serum to either support qPCR results. This affects major conclusion of the article and therefore should be thoroughly addressed.
5. Discussion is very limited as it just restates the results. No critical discussion in regard to the existing literature and importance to the filed is provided. Study limitations are not discussed.
Author Response
Reply to Reviewers Comments
The authors would like to express sincere thanks to the reviewers of Vaccines journal for vital comments on this manuscript for further improvement. As per the reviewer’s comments, the manuscript has been modified and the changes are presented in track-change mode. All the co-authors have participated in the revision of the manuscript.
Reviewer: 2
Comments to the Author (blind)
Major:
- The premise of the testing these many different cell lines is not clearly articulated. Is it needed for neutralization assays? Drug testing? What was done before? How is this study compared to the previously done in the field and cell lines that are commonly used in the field?
Reply:
- Our published research paper described the difficulties in isolation of omicron variants from human clinical samples, a combination of both in vivo and in vitro approaches were used. The difficulties in omicron isolation, necessitated to search for cell lines from diverse tissue and animal origins then available in our laboratory for rapid and easier isolation of omicron variant required for further research like omicron based tissue culture based omicron booster vaccine, evaluating/ re-evaluating the utility of different cell lines for standardization of assays such as neutralization assay, antiviral assays, etc in the context of emerging omicron variants.
- The main objective of the study was to identify highly permissive cell lines for omicron replication so as to produce virus in bulk for future omicron tissue culture based booster vaccines.
- Recently, there was an increase in the COVID-19 cases with Omicron variants which has greater transmissibility and vaccine efficacy remarkably dropped. Although several publications has compared the sensitivity of cell lines to SARS-CoV-2; there is need to study the susceptibility of these cell lines with Omicron variants.
- If authors did not test Wuhan strain on these cell lines, it will be beneficial to add appropriate citations to the Table 1 and create an additional column so readers can find relevant information in the literature. If authors tested Wuhan strain on some or all of these cell lines, it will very useful to include that information also. Otherwise, there is an impression that some of the cell lines that authors tested in the study very reported susceptible to Wuhan strain.
Reply:
- The main objective of the study was to check susceptibility of cell lines to omicron BA.1.1 infection.
- We have not tested susceptibility of cell lines against Wuhan strain.
- Appropriate citations related to Wuhan strain has been added in the Table 1 by creating additional column.
- Approach to define permissiveness is questionable. First of all, it is defined much later in the text then it is actually used. Then, change by 0.187 units defines permissive vs highly permissive but 4.6-0.15 is still permissive although it is logical to assign 0.15 as lowly permissive.
Reply:
- We have defined permissiveness based on three parameters viz. CPE, viral load and TCID50 on Vero CCL-81 cells as viral loads of fold changes of >0.15 in addition to TCID50 titer in Vero CCL-81 cells.
- Some corrections as suggested by the reviewer has been added in the revised manuscript.
- The results are thoroughly reanalyzed to draw concise conclusive remarks.
- Defining A549, RD and MRC5 as non-permissive is highly questionable. Obviously, there was no CPE on these cell lines, but virus was detected and it was infectious virus es evident by the productive P2 round. Moreover, differences in Baseline vs P1 and P1 vs P2 suggest that P2 was much more productive than P1 as there was not much less virus harvested after P2 compared to what was inoculated while during P1 the drop was significant. Therefore, qPCR indicates that these cell lines permitted virus infection without CPE. Because TCID50 measurement is based on CPE it yields obviously 0. To verify productive infection, it can be suggested to perform immunostaining of infected cells with antibody or serum to either support qPCR results. This affects major conclusion of the article and therefore should be thoroughly addressed.
Reply:
- The authors are in agreement with the reviewer. These cells are permissive as evident by the viral load of the both the passages. However, these cell lines could not be used for variety of reaserch purposes as there could not be known viral titre.
- With increasing work load in our laboratory realted to emerging SARS-CoV-2 variants and recent Monkeypox outbreak, it would be difficult for us to carry the immunostaining assay.
- Discussion is very limited as it just restates the results. No critical discussion in regard to the existing literature and importance to the filed is provided. Study limitations are not discussed.
Reply:
- The discussion has been revised.
- Study limitation have been included in the revised manuscript.
Minor:
- Editorial help with typos and formal English is warranted. Few examples: line 107 should read “on day 5”, line 109 “Biosafety level 4” no dashes, line 113: African green monkey embryo with clone MA-104; line 131: 5% CO2 and 75-80% humidity in a humidified CO2 incubator; line 154-155, word “using’ is used 3 times in one sentence.
Reply:
- The typographical errors, grammatical mistakes have been rectified in the revised manuscript as suggested by the reviewers.
- Line 175: not sure what this statement means: “4+ CPE (75-100% cells showing CPE) was observed with enhanced CPE”
Reply:
- Sentence has been reframed in the revised manuscript as suggested by the reviewer.
- What was the exact difference in CPE between Vero CCL-81, Vero E6 and Vero/ hSLAM and other cells? “Vero CCL-81, Vero E6 and Vero/ hSLAM cells showed distinct morphological changes with rounding and sloughing of cells throughout the cell culture flask” but the following text is confusing as it uses the same term to describe CPE for other cells.
Reply:
- The CPE description in each cell line is clearly explained in the revised manuscript.
- Figure 2 is redundant to the Table 3. Can be omitted.
Reply:
- Figure 2 omitted as suggested by the reviewer.
Round 2
Reviewer 1 Report
This is a well-written manuscript, but my two main major concerns remain after revision.
The article compares several cell lines, comparisons of which have not yet been published. As a conclusion, however, authors recommend the use of cell lines currently already used to test SARS-Cov2 and its omicron variant.
In the absence of statistical analysis, the suggestions made in the article are questionable. I recommend performing at least three parallel, independent series of experiments.
Author Response
Reply to Comments of Reviewer 1
Comments to the Author (blind)
- This is a well-written manuscript, but my two main major concerns remain after revision.
Reply: The authors would like to express sincere thanks to the reviewer 1 for vital comments on this manuscript for further improvement. As per the reviewer’s comments, the manuscript has been modified and the changes are presented in track-change mode. All the co-authors have participated in the revision of the manuscript.
- The article compares several cell lines, comparisons of which have not yet been published. As a conclusion, however, authors recommend the use of cell lines currently already used to test SARS-Cov2 and its omicron variant.
Reply: As per the important suggestion of the reviwer, the statement recommending the use of cell lines has been removed from the revised manuscript.
- In the absence of statistical analysis, the suggestions made in the article are questionable. I recommend performing at least three parallel, independent series of experiments.
Reply: As per the recommendation of the reviewer, the statistical analysis of the TCID50 values and viral loads of both P1 and P2 infection was carried out for all cell lines. The revision discussing the statistical analysis has been included in the revised manuscript.
Reviewer 2 Report
Manuscript has been throughly revised and all raised concerns have been properly addressed.
Thank you!
Author Response
Reply to Comments of Reviewer 2
- Manuscript has been thoroughly revised and all raised concerns have been properly addressed.
Reply: The authors would like to express sincere thanks to the reviewer 2 for vital comments on this manuscript for further improvement.
Round 3
Reviewer 1 Report
The aim of the article is to determine which cell line can be useful for in vitro studies of the SARS-CoV-2 varianst. I don't see the point of the combined examination of different cell lines of monkey origin. „The comparison of TCID 50 valuses in P1 and P2 infection found to be insignificant for monkey cell lines (p value = 0.18)” (If I understand correctly, the average of the P1 values of the 5 monkey cell lines is compared to the average of P2 value of the 5 cell lines.) The statistical analysis should refer to the comparison of individual cell lines.
Without performing three independent experiments, I do not consider the results reliable. In my opinion, the results cannot be published without repeating the experiments.
Author Response
Reply to Comments of Reviewer 1
Comments to the Author (blind)
- The aim of the article is to determine which cell line can be useful for in vitro studies of the SARS-CoV-2 variants. I don't see the point of the combined examination of different cell lines of monkey origin. The comparison of TCID 50 valuses in P1 and P2 infection found to be insignificant for monkey cell lines (p value = 0.18)” (If I understand correctly, the average of the P1 values of the 5 monkey cell lines is compared to the average of P2 value of the 5 cell lines.) The statistical analysis should refer to the comparison of individual cell lines.
Reply: Considering the two replicates of the experiment, the statistical comparison of individual cell lines can’t be statistically analyzed.
- Without performing three independent experiments, I do not consider the results reliable. In my opinion, the results cannot be published without repeating the experiments.
Reply: How many times to replicate an experiment is actually a design decision. Generally, the more it is replicated, the more accurate the results of the experiment will be. However, resources tend to be limited, which places constraints on the number of replications. In our study, Considering the number of cell lines and lengthy experiments, we utilize two replicates for the assays which are reproducible.